**Comment**

# When distrust is misplaced, social and democratic bonds weaken

Louisa Estadieu & Markus Langer

Misplaced distrust leads citizens to dismiss legitimate leaders, experts, or groups. This distrust may be fueled by prejudice or misinformation, and political rhetoric. This comment highlights the moral and democratic consequences of misplaced distrust.

Trust is often described as a "social glue" that is key to cooperation, collective action, and civic engagement. Less attention has been paid to its moral and normative implications. Trust is not only about anticipating others' behavior, but also about assessing them as legitimate participants in shared social and political life. When trust is withdrawn, it does more than signal caution: it questions others' credibility and (moral) standing, influencing how they are perceived and treated in collective activities.

One important case in which trust assessments go wrong is misplaced distrust. Misplaced distrust refers to situations in which someone or groups are distrusted based on prejudice, biases, or misinformation rather than evidence[1]. We argue that such misplacements fuel social fragmentation and contribute to a broader erosion of trust in institutions and society at large.

## The psychological roots of misplaced distrust

Distrust arises from an interplay of cognitive, affective, and motivational processes. Individuals assess whether to trust others based on informational cues that may or may not reflect actual behavior or intentions[2]. Heuristics and cognitive biases, including in-group biases, shape these judgments. These judgments form quickly, drawing on shared norms, group membership, and previous experiences, and can make individuals suspicious of those perceived as different, even when individuals attempt to avoid prejudice[3].

Motivational and identity-protective reasoning further amplifies distrust. Individuals often avoid engaging with, or dismiss information from, those challenging their core beliefs or social identity. These factors can also prevent people from even considering evidence that might contradict their initial distrust[4]. Such dynamics can undermine shared standards for evaluating trustworthiness, such as goodwill or accountability, and institutionalize distrust[5]. Notable examples of institutionalized distrust include the systematic underestimation of women's reported symptoms in medical settings and the undifferentiated distrust of racialized groups in political rhetoric[6].

## Misplaced distrust operates across societal levels

Misplaced distrust includes cases where perceptions and actual trustworthiness diverge. The underlying drivers may operate at the individual level (e.g., cognitive biases, identity-protective reasoning), the system level (e.g., misinformation campaigns, political rhetoric), or both. In all cases, such distrust harms and wrongs individuals or groups by limiting opportunities for those who are (unjustifiably) distrusted.

The misplacement of distrust takes many forms. It spreads laterally when leaders undermine one another. It also moves vertically: Top-down, politicians may generalize distrust toward segments of the public, while bottom-up, citizens may dismiss politicians or parties despite evidence of trustworthiness, such as competence, integrity, transparency, or benevolence towards citizens. These tendencies are reinforced by social-network dynamics: individuals often rely on information from sources they already consider trustworthy and communicate their evaluations within the same networks.

However, evaluating the misplacement of distrust is not always easy, as it relies on both normative standards, such as democratic values, and empirical evidence, which provide benchmarks for evaluating the justification and scope of distrust in particular situations.

## Consequences of misplaced distrust

**Behavioral and individual-level consequences.** Reciprocal interactions are what build and sustain trust, yet misplaced distrust undermines this equilibrium. People become less willing to engage with those they distrust, which reduces their opportunities to revise their initial judgment and makes it harder for the respective other to improve their image. Also, those who express distrust can inadvertently send cues that others rely on when forming judgments about trustworthiness. This generates self-reinforcing cycles in which agents are repeatedly labeled untrustworthy, and such assessments are shared with others. As a consequence, both distrusted individuals and distrusting individuals isolate themselves, and suspicion grows: ambiguous actions are interpreted as confirmation of distrust, while signs of reliability or goodwill are often overlooked.

**Epistemic and moral consequences.** Distrust is not inherently harmful. When justified, it is a constitutive attitude that enables critique and resistance to abuses of power. However, if distrust is misplaced, it can cause both moral and epistemic harm of those involved by producing exclusionary effects. Individuals or groups who are unjustifiably distrusted are less likely to be heard, less likely to be believed, and less likely to be included in decision-making processes[7]. Over time, this dynamic can push those persistently distrusted to withdraw from civic and epistemic participation. At the same time, those who express distrust are affected as well: by avoiding engagement with the distrusted parties, they also reduce their own opportunities for collaboration and learning.

**Collective consequences.** As with individuals, ongoing exposure to distrust reduces political and civic participation and engagement among groups that are unjustifiably distrusted or perceive themselves as such[8]. In some cases, misplaced distrust can also motivate activism (or even extremism) rather than disengagement.

As misplaced distrust sets a climate in which groups interpret one another's intentions through suspicion rather than recognition, shared norms, including democratic ones, come under pressure. For example, when politicians or institutions signal unspecified distrust toward specific social groups, they delegitimize these groups and reduce their perceived social standing. Members of these groups may then reciprocate with distrust towards norms of institutions and political actors[9]. At the same time, citizens may also unjustifiably distrust political actors despite evidence of competence or integrity. These dynamics can lead to intensified political and social polarization.

Such patterns also contribute to a systematic miscalibration of trust, with some actors being overly trusted despite warning signals, while others are distrusted despite evidence. As a result, individuals may struggle to make accurate trust judgments based on evidence and clear normative standards, and social divisions are intensified as both people and institutions adjust their behavior according to perceived rather than actual trustworthiness.

Many call for more trust in institutions, parties, and leaders. However, the problem is not distrust per se, but its misplacement and a lack of calibrated trust. Simply aiming for "more trust" may be the reason for polarization because some trust one side of the political spectrum and others the other side—but none do this in a calibrated way.

**New pathways in addressing misplaced distrust**. The task ahead is thus not only to understand why trust has eroded in the first place but also to understand how to rebuild mutual and calibrated (dis)trust. Central to this effort is breaking the self-reinforcing cycle of misplaced distrust, where distrust breeds further distrust and undermines social cohesion. So how do we move forward?

First, precise definitions of distrust are key to both understanding and measurement. While trust has been extensively studied, distrust remains comparatively underexplored as a distinct phenomenon. Without clear distinctions between trust and distrust and related notions such as mistrust, suspicion, or skepticism, or between different forms, such as misplaced versus well-placed, research risks conflating fundamentally different processes. Crucially, these distinctions should also make explicit the normative assumptions underlying distrust, including how judgments and perceptions of trustworthiness and untrustworthiness are formed. Such assumptions are often implicit, and making them explicit is key to avoiding superficial or misleading conclusions. Precision also helps to move the conversation beyond simplistic calls for "more trust" toward context-sensitive, evidence-based evaluations and comparable measurements.

Building on this, political institutions and citizens should articulate clearer normative benchmarks for evaluating trust and distrust. Many treat distrust as inherently problematic without specifying the standards that define what is considered trustworthy. Greater effort should go into making these standards explicit. Political institutions need to clarify what forms of trustworthiness they want to embody and what criteria they expect citizens to use when assessing them. At the same time, citizens need to be clear about what exactly they consider trustworthy, and they should, in turn, act accordingly. Integrating both top-down (institutional definitions) and bottom-up (citizen assessments) perspectives might help to re-establish mutual trust.

Third, misplaced distrust is experienced differently across groups, and its consequences vary with context and community. Although large-scale surveys, such as those conducted by the OECD, provide valuable insight into the current decline in trust in political institutions, more research is needed to understand the specific community- and context-level factors driving these declines and the strategies required to address misplaced distrust. Future research should examine in more depth how misplaced distrust is perceived and moralized within different groups, and how it shapes the formation of alternative trust networks. Funding bodies should similarly support large-scale interventions to test mechanisms that encourage how to initiate reengagement in trust relationships, as well as restart trustworthiness assessment of mistrusted groups.

Finally, prevention requires active engagement from institutions and science. Building on important work on how trust can be rebuilt through transparency and accountability, many initiatives have already provided valuable guidance[10]. Transparency is not an end in itself: it should support citizens in assessing trustworthiness and achieving calibrated trust—simply putting out information on government websites in 100+ page documents will likely not help achieve these goals. Addressing this requires looking beyond bottom-up flows of trust from citizens to institutions, which dominate current research, and understanding how top-down (dis)trust dynamics lead institutions to extend, withhold, or withdraw goal-directed transparency.

Sustaining trust over time requires ongoing reflection on norms and standards for justified (dis)trust. Clearly defining the criteria of trust and distrust helps societies break self-reinforcing cycles of misplaced distrust.

**Louisa Estadieu** [ID][1] [✉] **& Markus Langer** [ID][2]

[1]Department of Humanities, Social and Political Sciences, ETH Zürich, Zürich, Switzerland. [2]Department of Psychology, University of Freiburg, Freiburg, Germany. [✉]e-mail: louisa.estadieu@gess.ethz.ch

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

### Acknowledgements
We thank Nadia Mazouz, Viktoria Cologna, and the members of the Practical Philosophy group at ETH Zürich for helpful feedback on the comment.

## Author contributions
L.E. conceived the article idea and prepared the original draft of the manuscript. M.L. contributed to developing the concept and revising the manuscript. Both authors collaborated on incorporating reviewer comments and participated in the final editing and approval of the manuscript.

## Competing interests
The authors declare no competing interests.

## Additional information

**Peer review information** The manuscript was considered suitable for publication without further review at Communications Psychology. Primary Handling Editor: Marike Schiffer. A peer review file is available.

