## [Peer Review file · Communications Psychology]

When distrust is misplaced, social and democratic bonds weaken

Corresponding Author: Dr Louisa Estadieu

Version 0:

Decision Letter:

Dear Dr Estadieu,

Thank you for submitting your manuscript titled "Misplaced Distrust" to Communications Psychology. We have given the paper our careful consideration and find it of potential interest. However, due to certain shortcomings we cannot consider it further in its present state.

Your Comment posits that misplaced distrust is detrimental to society and democracies, and briefly mentions three areas to which this applies, as well as listing a few future research questions. The piece focuses strongly on the societal and institutional level; it is conceptually informed by moral ethics and presents a viewpoint akin to that of sociology or political science.

Communications Psychology is a journal for psychologists and scientists in related disciplines that work on the same research questions. To be suitable as a Comment in the journal, the piece would need to take a more individual-level perspective to break down what psychological processes underpin abstract processes. The list of consequences of misplaced distrust which you describe at the collective level would need to be expanded, both in terms of the space they take within the piece (and level of explanation), as well as in scope to include individual-level effects or a more psychology-informed narrative.

The list of research questions would likewise need to be developed further, to balance the piece more towards a forward-looking commentary that includes compelling recommendations for the fields of (political or social) psychology.

We shall hope to receive your revised version as soon as you are able to complete the suggested revisions. If something similar is published in the interim we will have to consider the impact it has on the novelty of a revised manuscript.

If you anticipate a delay of more than four weeks, please let us know. If you are not interested in submitting a suitably revised manuscript in the future please let me know immediately so we can close your file. If you have any questions, please contact me.

Please use the link below when you are prepared to resubmit.

Link Redacted

Thank you for your interest in Communications Psychology.

Best regards,
Marika

Marika Schiffer, PhD
Chief Editor
Communications Psychology

Version 1:

Decision Letter:

Dear Dr Estadieu,

Thank you for your patience during the editorial evaluation of your revised Comment. I have now had the time to read your work carefully. We remain very interested in publishing your Comment and I recognize the significant revisions you have undertaken to make the work more suitable for the journal.

At this stage, there are a number of high-level issues and more detailed edits that need to be addressed to make the work suitable for publication.

To aid you with that task, I have included a marked-up version of your manuscript.

On a more general level, while the focus on psychology is developed better, the final section in which the forward-looking aspects are covered, requires more work. Comments are forward-looking pieces and that needs to be reflected in the balancing of the piece sections, with the outlook taking up about a third of the piece. The flow of arguments and order of paragraphs also needs to be revisited to tighten the narrative towards this end, and to avoid redundancies.

On a more detailed level, the language is sometimes overly complicated, which makes it harder for the reader to follow the argument. One general recommendation is to avoid introducing any new or unusual terminology. The second is to state arguments only once, rather than presenting multiple versions of the same argument in immediate succession.

EDITORIAL POLICIES AND FORMATTING

You will find a complete list of formatting requirements following this link: <https://www.nature.com/documents/commsj-style-formatting-checklist-review-perspective.pdf>

Please use the checklist to prepare your manuscript for resubmission.

*** TRANSPARENT PEER REVIEW:** Communications Psychology uses a transparent peer review system. This means that we publish the editorial decision letters including Reviewers' comments to the authors and the author rebuttal letters online as a supplementary peer review file. We publish these records for all accepted manuscripts. However, on author request, confidential information and data can be removed from the published reviewer reports and rebuttal letters prior to publication. If your manuscript has been previously reviewed at another journal, those Reviewers' comments would not form part of the published peer review file.

If you have any questions about any of our policies or formatting, please don't hesitate to contact me.

Please use the following link to submit your revised manuscript:

Link Redacted

We hope to receive your revised paper within 12 weeks; please let us know if you aren't able to submit it within this time so that we can discuss how best to proceed. If we don't hear from you, and the revision process takes significantly longer, we may close your file.

Please do not hesitate to contact me if you have any questions or would like to discuss these revisions further. We look forward to seeing the revised manuscript and thank you for the opportunity to review your work.

Best regards,

Marika

Marika Schiffer, PhD
Chief Editor
Communications Psychology

**** Visit Nature Research's author and referees' website at www.nature.com/authors for information about policies, services and author benefits****

Version 2:

Decision Letter:

**** Please ensure you delete the link to your author homepage in this e-mail if you wish to forward it to your co-authors ****

Dear Dr Estadieu,

Your Comment titled "When distrust is misplaced, social and democratic bonds weaken" has now been editorially evaluated. I am delighted to say that we are happy, in principle, to publish it in Communications Psychology.

If the revised paper is in Communications Psychology format, in an accessible style, and of appropriate length, we shall accept it for publication immediately.

*If you have not done so already, please alert me to any related manuscripts from your group that are under consideration or in press at other journals, or are being written up for submission to other journals (see www.nature.com/authors/editorial_policies/duplicate.html for details).

FORMATTING GUIDELINES:

Please use the attached checklist to prepare your manuscript for final submission.

Communications Psychology is a fully open access journal. Articles are made freely accessible on publication. For further information about article processing charges, open access funding, and advice and support from Nature Research, please visit <https://www.nature.com/commpsychol/open-access>.

At acceptance, you will be provided with instructions for completing the open access licence agreement on behalf of all authors. This grants us the necessary permissions to publish your paper. Additionally, you will be asked to declare that all required third party permissions have been obtained.

Please note that your paper cannot be sent for typesetting to our production team until we have received this information; **therefore, please ensure that you have this ready when submitting the final version of your manuscript.**

ORCID

Communications Psychology is committed to improving transparency in authorship. As part of our efforts in this direction, we are now requesting that all authors identified as 'corresponding author' create and link their Open Researcher and Contributor Identifier (ORCID) with their account on the Manuscript Tracking System (MTS) prior to acceptance. ORCID helps the scientific community achieve unambiguous attribution of all scholarly contributions. For more information please visit <http://www.springernature.com/orcid>

For all corresponding authors listed on the manuscript, please follow the instructions in the link below to link your ORCID to your account on our MTS before submitting the final version of the manuscript. If you do not yet have an ORCID you will be able to create one in minutes.

IMPORTANT: All authors identified as 'corresponding author' on the manuscript must follow these instructions. Non-corresponding authors do not have to link their ORCIDs but are encouraged to do so. Please note that it will not be possible to add/modify ORCIDs at proof. Thus, if they wish to have their ORCID added to the paper they must also follow the above procedure prior to acceptance.

To support ORCID's aims, we only allow a single ORCID identifier to be attached to one account. If you have any issues attaching an ORCID identifier to your MTS account, please contact the [Platform Support Helpdesk](http://platformsupport.nature.com/).

Support Helpdesk.

Link Redacted

We hope to hear from you within two weeks; please let us know if the process may take longer.

Best regards,

Marike Schiffer

Marike Schiffer, PhD
Chief Editor
Communications Psychology
